# First Report of *aac(6′)-Ib* and *aac(6′)-Ib-cr* Variant Genes Associated with Mutations in *gyrA* Encoded Fluoroquinolone Resistance in Avian *Campylobacter coli* Strains Collected in Tunisia

**DOI:** 10.3390/ijms242216116

**Published:** 2023-11-09

**Authors:** Manel Gharbi, Mohammed Abdo Saghir Abbas, Safa Hamrouni, Abderrazak Maaroufi

**Affiliations:** 1Group of Bacteriology and Biotechnology Development, Laboratory of Epidemiology and Veterinary Microbiology, Institut Pasteur de Tunis, University of Tunis El Manar (UTM), Tunis 1002, Tunisia; safa.hamrouni@pasteur.tn (S.H.); abderrazak.maaroufi@pasteur.tn (A.M.); 2Unit of Vector Ecology, Pasteur Institute of Tunis, Tunis 1002, Tunisia; m.abbas9900@gmail.com; 3University of Tunis El Manar (UTM), Tunis 1002, Tunisia

**Keywords:** *Campylobacter*, poultry, fluoroquinolone-resistance, phylogeny, *aac(6′)-Ib-cr*

## Abstract

The *aac(6′)-Ib* gene is the most widespread gene encoding aminoglycoside-modifying enzyme and conferring resistance to tobramycin, streptomycin and kanamycin. The variant *aac(6′)-Ib-cr* gene confers resistance to both aminoglycosides and fluoroquinolones (FQ). A total of 132 *Campylobacter* isolates, including 91 *C. jejuni* and 41 *C. coli*, were selected from broiler hens isolates. The *aac(6′)-Ib* gene was amplified using PCR and was subsequently digested with the *BtsCI* restriction enzyme to identify *aac(6′)-Ib-cr*. Among these isolates, 31 out of 41 *C. coli* (75.6%) and 1 (0.98%) *C. jejuni* were positive for the *aac(6′)*-Ib gene, which was identified as the *aac(6′)-Ib-cr* variant in 10 (32.25%) *C. coli* isolates. This variant was correlated with mutations in gyrA (Thr-86-Ile), as well as resistance to FQs. This study is the first report in Tunisia on *Campylobacter coli* strains harboring both the *aac(6′)-Ib* and *aac(6′)-Ib-cr* variants. These genes were present in *Campylobacter* isolates exhibiting resistance to multiple antibiotics, which restricts the range of available treatments.

## 1. Introduction

*Campylobacter* is an important zoonotic pathogen, and campylobacteriosis constitutes a public health problem worldwide. Fluoroquinolones are one of the alternative treatments for invasive infections caused by *Campylobacter* in adult patients. The emergence of antimicrobial resistance involves some therapeutic failures or restrictions in the use of antimicrobial agents. However, high-level fluoroquinolone resistance is relatively uncommon in *Campylobacter*, probably due to the prohibitive fitness cost of the bacteria [1,2]. Quinolone/fluoroquinolone resistance is mainly caused by chromosomal mutations in the genes that control the production of efflux pumps or bacterial topoisomerases (DNA gyrase and topoisomerase IV), or both [3]. However, transferable mechanisms known as plasmid-mediated quinolone resistance (PMQR) have emerged during recent decades especially in Gram-negative bacteria [4]. PMQR determinants are commonly reported in fluoroquinolone-resistant *Enterobacteriaceae* and include *qnr* (*qnrA*, *qnrB*, *qnrC*, *qnrD*, *qnrS* and *qnrVC*), *qep*, *oqxAB* or *aac(6′)-Ib-cr* genes [5]. The aminoglycoside acetyltransferase AAC(6′)-Ib enzyme, which confers resistance to tobramycin, kanamycin, and amikacin, was first identified in *Klebsiella pneumoniae* isolates in 1986. Since then, several variants of this enzyme have been described [6,7]. The *aac(6′)-Ib-cr* variant of *aac(6′)-Ib* encodes an aminoglycoside acetyltransferase that contributes reduced susceptibility to ciprofloxacin via N-acetylation of its piperazinyl amine [8]. Thus, AAC(6′)-Ib-cr is a bifunctional acetyltransferase, capable of modifying both aminoglycosides and quinolones. The AAC(6′)-Ib-cr enzyme contains two amino acid modifications, Asp179Tyr and Trp102Arg [9]. Globally, this enzyme is widely reported, either plasmid or chromosomally encoded, from Gram-negative bacteria, especially in Enterobacteriaceae of human, farm animal and environmental origins, showing its great potential to be disseminated [10]. In general, in bacteria, the spread of genes encoding antimicrobial-resistance is mediated by mobile genetic elements such as plasmids, transposons and insertion sequences as well as integrons, and rarely by phages, which enhance their rapid and large-scale spread [11].

Fluoroquinolones are among the three most widely used antimicrobials (macrolides and tetracyclines) used to treat systemic infection caused by *Campylobacter*; in addition, gentamicin (an aminoglycoside) is also used in some cases [1,2]. Taken together, it is crucial to identify the strains harboring the *aac(6′)-Ib-cr* variant in order to employ fluoroquinolone antimicrobials appropriately in human therapy. The aim of this study is to present the characteristics of campylobacter isolates from poultry origin harboring the *aac(6′)-Ib-cr* variant. This study, the first of its kind in Tunisia, will be useful for the future studies of emerging genes conferring antibiotic resistance.

## 2. Results

### 2.1. Antimicrobial Susceptibility Testing

A total of 132 isolates was recovered from cloacal swabs from poultry, including 91 *C. jejuni* and 41 *C. coli* characterized as multi-drug-resistant, of which (99.2%) were resistant to ciprofloxacin and 46.2% to nalidixic acid. To a lesser extent, the resistance rates of aminoglycoside were about 12.9% to gentamicin, 49% to tobramycin, 51.4% to streptomycin, and 52% to kanamycin. The resistance rates of *C. jejuni* and *C. coli* isolates to kanamycin (n = 19.78% vs. 46, 34%), tobramycin (19.78% vs. 60, 97%), and NA (57.1% versus 22%) differ significantly (*p* < 0.05). For the other aminoglycosides (gentamicin and streptomycin), there were no significant differences between both species (Table 1).

### 2.2. Detection of Mutations in GyrA

Nearly 90% of *C. jejuni* and 80% of *C. coli* isolates that were resistant to ciprofloxacin and/or nalidixic acid had the Thr-86-Ile amino acid substitution in their GyrA, as determined using MAMA-PCR. The Thr-86-Ile alteration was not found in one *C. jejuni* isolate that was susceptible to both ciprofloxacin and/or nalidixic acid.

### 2.3. Prevalence of the aac(6′)-Ib and aac(6′)-Ib-cr Genes

PCR of the *aac(6*′*)-Ib* gene showed that 31/41 (75.60%) *C. coli* isolates and 1/91 (0.98%) *C. jejuni* isolates were positive for the *aac*(*6′)-Ib* gene. Following digestion of the amplified amplicons with *BtsCI*, wild-type *aac(6′)-Ib* revealed two DNA fragments (272 and 210 bp) on gel electrophoresis (Figure 1). Interestingly, 10 *C. coli* isolates showed three fragments (482 bp, 272 bp, 210 bp) of DNA on gel electrophoresis when treated with the restriction enzyme *BtsCI*, indicating the occurrence of both the wild-type *aac(6′)-Ib* gene and the *aac(6′)-Ib-cr* variant (Figure 1).

### 2.4. aac(6′)-Ib Sequencing and Phylogenetic Analysis

Among the isolates harboring the *aac(6′)-Ib* gene based on their antibiotic resistance profiles and *BtsCI* digestion results, the *aac(6′)-Ib* of one *C. jejuni* isolate and four *C. coli* isolates (one harboring the *aac(6′)-Ib-cr*) was sequenced. The sequences of the *aac(6′)-Ib* gene were deposited in GenBank under accession numbers OR124788, OR125068, OR125070, OR125069, and OR192836.

A phylogenetic analysis was performed to observe the phylogenetic relationships of *aac(6′)-Ib* gene among other bacterial species. The sequence analysis showed that Tunisian *aac(6′)-Ib* sequences found in this study (OR124788, OR125068, OR125070, OR125069, and OR192836) were closely related to other sequences from various species, such as *Enterobacter cloacae*, *Citrobacter freundii*, and *Pseudomonadaceae*, and the phylogenetic branch was supported by a high bootstrap value of 99% (Figure 2). Moreover, clustering using the maximum likelihood method defined four sequence clusters and one singleton corresponding to *C. jejuni* AF4339785 (*aacA4*) (Figure 2). Different clusters can exhibit similar properties, while others show substantial differences in their characteristics.

## 3. Discussion

In order to counter the selection pressure brought on by the use of antibiotics in both veterinary and human medicine, *Campylobacter* has developed a variety of molecular mechanisms of antibiotic resistance, such as modification or mutation of antimicrobial targets, modification or inactivation of antibiotics, and decreased drug accumulation by drug efflux pumps.

According to our findings, the rate of ciprofloxacin resistance (99.2%) in *Campylobacter* isolates is quite high. The fact that the same antimicrobials are frequently used as first-line treatments in broiler flocks to combat bacterial illnesses can be blamed for the high rate of resistance to ciprofloxacin, which is equivalent to that found in Italy [12,13,14]. Several studies have unequivocally demonstrated a link between the use of fuoroquinolones in chicken production and a rise in the resistance of *Campylobacter* isolates of avian and human origins [15]. When considered together, the excessive and unreasonable use of antimicrobial medications is clearly related to the high rates of antibiotic resistance. There are two different mechanisms that contribute to ciprofloxacin resistance. The first is the presence of the *cmeABC* operon, the most common *Campylobacter* multidrug efflux pump, which contributes to fluoroquinolone resistance by reducing the amount of drug in cells [16,17]. Amino acid substitutions in the quinolone resistance-determining region (QRDR) of the *gyrA* gene, particularly the Thr-86-Ile mutation, common in fluoroquinolone-resistant isolates, is the second mechanism of resistance [18,19]. This mutation was found in 90% and 80% of quinolone-resistant *C. jejuni* and *C. coli* isolates, respectively. Since the remaining resistant isolates have the *cmeB* gene and do not have the Thr-86-Ile substitution, as reported in our previous work on these isolates [20], the observed resistance may be due to the action of an efflux pump or to the presence of additional alterations in the ParC subunit of topoisomerase IV.

On the other hand, we showed high rates of resistance to kanamycin (28.03%), tobramycin (58.33%), and streptomycin (48.48%), which are comparable to rates reported in the USA and China [20]. In fact, high levels of aminoglycoside resistance are due to the use of these antimicrobial drugs in recent years in humans and their administration to treat occasional systemic infections caused by *Campylobacter*; the emergence and spread of these resistance markers represent a potential concern for public health, especially concerning gentamicin resistance, which is an antibiotic occasionally used in systemic infection caused by *Campylobacter* isolates [21].

Notably, new antibiotic resistance mechanisms continuously emerge in *Campylobacter*. Aminoglycoside phosphotransferases constitute the majority of aminoglycoside-modifying enzymes identified in *Campylobacter* spp. They mediate kanamycin and neomycin resistance. The AAC(6′)-Ib is a clinically significant enzyme commonly present in a number of Gram-negative bacteria. It confers resistance to tobramycin, kanamycin, and amikacin. The AAC(6′)-Ib enzyme is interesting not only because of its widespread distribution, but also because the *aac(6)-Ib* gene is frequently located on integrons, transposons, plasmids, genomic islands, and other genetic structures. In addition, the *aac(6′)-Ib-cr* mutant gene encodes resistance to both fluoroquinolones and aminoglycosides. The frequency of *aac(6′)-Ib-cr* found in this study exceeds those found in previous studies in Brazil (up to 4%) in clinical *Pseudomonas aeruginosa* and *Providencia stuartii* isolates [22,23] and from samples obtained in different countries, which ranged from 12% to 23.3% [24,25]. Notably, the use of broad-spectrum bactericidal aminoglycosides routinely in children, primarily for infections caused by Gram-negative fluoroquinolone-resistant pathogens, has increased worldwide [26]. Therefore, this use of aminoglycosides would have increased the likelihood occurrence of various transferable genes encoding resistance against them including wild-type *aac(6′)-Ib*. Consequently, the emergence of the *aac(6′)-Ib-cr* variant could be the result of an enormous number of silent mutations in this wild gene, of which one has acquired the advantage of being a bi-functional enzyme, the ‘cr” variant. In addition, the worry for this variant is that it substantially enhanced the frequency of selection of chromosomal mutations in DNA gyrase and topoisomerase IV upon exposure to ciprofloxacin [27,28].

In our isolates, the *aac(6′)-Ib-cr* variant among those harboring *aac(6′)-Ib* was extremely high (31.25%), and only identified among *C. coli* isolates. This rate was much higher than those reported *E. coli* and other *Enterobactereaceae* isolates in other countries (4–15.6%) [29,30,31]. However, among clinical *E. coli* isolates collected in Shanghai, China, in 2000 to 2001, 51% had the ‘cr’ variant of *aac(6′)-Ib* [30]. As discussed above, all studies dealing with *aac(6′)-Ib-cr* were performed on isolates belonging to Gram-negative bacteria; however, to the best of our knowledge, only one study has reported the occurrence of an *aacA4* allele (an aminoglycoside 6′-*N*-acetyltransferase) from a *C. jejuni* strain isolated from a broiler chicken house environment, which demonstrated 100% homology to the AAC(6′)-Ib_7_ enzymes that confer characteristics resulting in higher kanamycin and tobramycin MICs, but only slightly increased resistance to gentamicin [32]. Therefore, we report for the first time the occurrence of *aac(6′)-Ib-cr* in *Campylobacter* isolates.

Furthermore, clustering using a maximum likelihood method based on the Kimura 2-parameter model was performed. Different clusters can exhibit similar properties, while others show substantial differences in their characteristics such as those cases in which there are significant specificity variations. The tree with the highest log likelihood (−1721.35) is shown. The percentage of trees in which the associated taxa clustered together is shown next to the branches. Initial tree(s) for the heuristic search were obtained automatically by applying Neighbor-Join and BioNJ algorithms to a matrix of pairwise distances estimated using the maximum composite likelihood (MCL) approach, and then selecting the topology with a superior log likelihood value. The analysis involved 34 nucleotide sequences. Phylogenetic analyses showed four sequence clusters and one singleton corresponding to *C. jejuni* AF4339785 (*aacA4*). The *aac(6′)-Ib* sequences of our five isolates (accession no: OR124788, OR125068, OR125070, OR125069, and OR192836) clustered with sequences derived from different species and countries. Using the information available, it is still not clear whether all aac(6′) enzymes evolved from a single origin, or whether the four groups are less related and the 6′acetylating activity has evolved independently at least three times [33]. According to the phylogenetic analyses reported by Salipante and Hall [33], the *aac(6′)-Ib* is most closely related to *aac(6′)-IIa*, *aac(6′)-IIb*, aac(6′)-IIc, and *aac(6′)-IId*.

## 4. Materials and Methods

### 4.1. Ethics Statement

Biomedical Ethics Committee of the ‘Institut Pasteur de Tunis’ approved this work, and the sample technique was carried out in accordance with the widely accepted ISO 10272-1:2006 Appendix 2 standards for the identification of *Campylobacter* spp.

### 4.2. Campylobacter Isolates Collection and Phenotypic Susceptibility

Herein, we tested 132 *Campylobacter* isolates, including 91 *C. jejuni* and 41 *C. coli.* These isolates were previously recovered from layer chickens [33] and tested against gentamicin (GEN, 10 μg), kanamycin (K, 30 μg), tobramycin, (TOB, 10 μg), streptomycin (SMN, 10 μg), nalidixic acid (NAL, 30 μg), and ciprofloxacin (CIP, 5 μg) (Bio-Rad, Marnes-la-Coquette, France) [34,35], using the disk diffusion method on Mueller–Hinton medium (Bio Life, Milan, Italy). Interpretation of antimicrobial susceptibility against each antibiotic was determined following the breakpoint values of the European Committee on Antimicrobial Susceptibility Testing (EUCAST) [35,36].

### 4.3. Extraction of Bacterial DNA

The genomic DNA of collected isolates was extracted using the boiling method [32]. Briefly, *Campylobacter* isolates were grown in 2 mL Bolton broth (Oxoid, United Kingdom) and plated on Karmali agar (SIGMA-Aldrich, Germany). *Campylobacter* colonies were then harvested and suspended in 100 μL TE buffer (10 mM Tris, 1 mM EDTA, pH 8.0). Cell suspensions were heated at 100 °C for 10 min and then cooled at −20 °C. Thereafter, cell suspensions were centrifuged at 8000 rpm for 5 min [32]. The supernatant containing DNA was collected, transferred into a new tube, and then stored at −20 °C until use.

### 4.4. Detection of Mutation in the QRDR of gyrA via Mismatch Amplification Mutation Assay (MAMA-PCR)

The quinolone region determining resistance (QRDR) of the *GyrA* subunit of the ADN gyrase enzyme is where the fluoroquinolone resistance is often codified by a point mutation (Thr-86-Ile) [37]. A single-point mutation Thr-86-Ile in the quinolone resistance-determining region (QRDR) of the *gyrA* gene was defined as the main mechanism of high-level resistance to fluoroquinolones. *Campylobacter* isolates were subjected to analysis via MAMA-PCR, as previously described for *C. jejuni* and *C. coli* isolates [37,38].

### 4.5. Détection and Sequencing of aac(6′)-Ib in Selected Isolates

*aac(6)-Ib* was amplified by PCR using primers 5′-TTGCGATGCTCTATGAGTGGCTA-3′ and 5′-CTCGAATGCCTGGCGTGTTT-3′ to produce a 482-bp amplicon. PCR conditions were 94 °C for 45 s, 55 °C for 45 s, and 72 °C for 45 s for 34 cycles. Positive and negative strains for *aac(6)-Ib* were included as controls. All PCR products positive for *aac(6’)-Ib* were further analyzed via digestion with *BtsCI* (New England Biolabs, England, UK).

Positive PCR products in selected five isolates were directly sequenced to identify the *aac(6)-Ib* gene. Nucleotide sequencing was carried out with an Applied Biosystems™ 3500 sequencer calibrated with BigDye^®^ Terminator V3.1. DNA sequences were assembled and edited using BioEdit Sequence Alignment Editor software version 7.0.5.3. The consensus sequences obtained were compared to sequences in the Basic Local Alignment Search tool (BLAST). The sequences generated in this study were deposited in GenBank under the following accession numbers: OR124788, OR125068, OR125070, OR125069, and OR192836.

### 4.6. Molecular Phylogenetic Analysis via the Maximum Likelihood Method

The evolutionary history was inferred using the maximum likelihood method based on the Kimura two-parameter model [39]. The tree with the highest log likelihood (−1721.35) is shown. The percentage of trees in which the associated taxa clustered together is shown next to the branches. Initial tree(s) for the heuristic search were obtained automatically by applying Neighbor-Join and BioNJ algorithms to a matrix of pairwise distances estimated using the maximum composite likelihood (MCL) approach, and then selecting the topology with a superior log likelihood value. A discrete Gamma distribution was used to model evolutionary rate differences among sites (five categories (+*G*, parameter = 200.0000)). The rate variation model allowed for some sites to be evolutionarily invariable ([+*I*], 8.19% sites). The tree is drawn to scale, with branch lengths measured in the number of substitutions per site. The analysis involved 34 nucleotide sequences. All positions containing gaps and missing data were eliminated. There were a total of 421 positions in the final dataset. Evolutionary analyses were conducted in MEGA7 [40].

### 4.7. Data Analysis

All collected data were analyzed using R software (www.infostat.com.ar/ accessed 24 July 2023). The antimicrobial resistance analyses were performed by means of a Chi-square statistic (*p* < 0.05) [41]. This test is a nonparametric tool designed to compare frequency counts between two groups of different sample sizes; the selection criterion for significantly prevalent variance was a stringent *p*-value of 0.001 or less. The phylogenetic tree was generated using iTOL version 6 (https://itol.embl.de/ accessed 5 September 2023). The strains of our study are reported using their accession number code displayed in bold.

## 5. Conclusions

The results of this study showed a significant prevalence of *aac(6′)-Ib* and *aac(6′)-Ib-cr*, as well as a high rate of co-carriage of these two genes among *Campylobacter* isolates in Tunisia. It seems that quinolone/fluoroquinolone resistance is established, mainly in *C. coli*, by chromosomal mutations the *gyrA* gene, and evolves to acquire *aac(6′)-Ib-cr* against a genetic background already rich in wild-type *aac(6′)-Ib* gene.

## Figures and Tables

**Figure 1 ijms-24-16116-f001:**
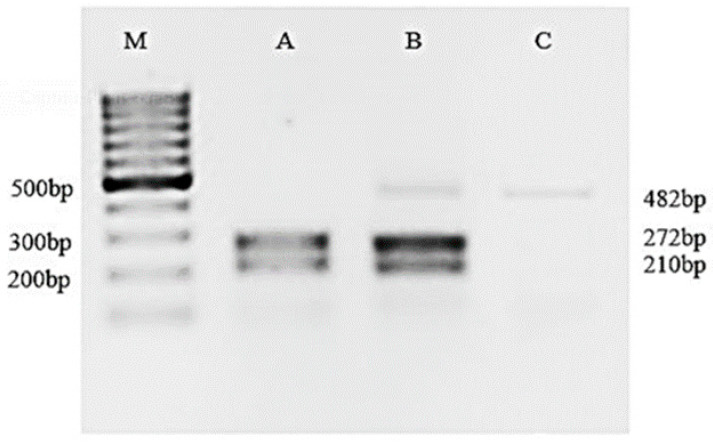
*BtsCI* digestion of purified *aac*(6′)-*Ib* PCR products. Lane M: 100 bp DNA ladder. Lane A: PCR products cut via *BtsCI* digestion (wild *aac(6′)-Ib* gene); lane B: a PCR product showing three fragments by *BtsCI* digestion (wild *aac(6′)-Ib*+*aac(6′)-Ib-cr* gene); lane C: undigested PCR product.

**Figure 2 ijms-24-16116-f002:**
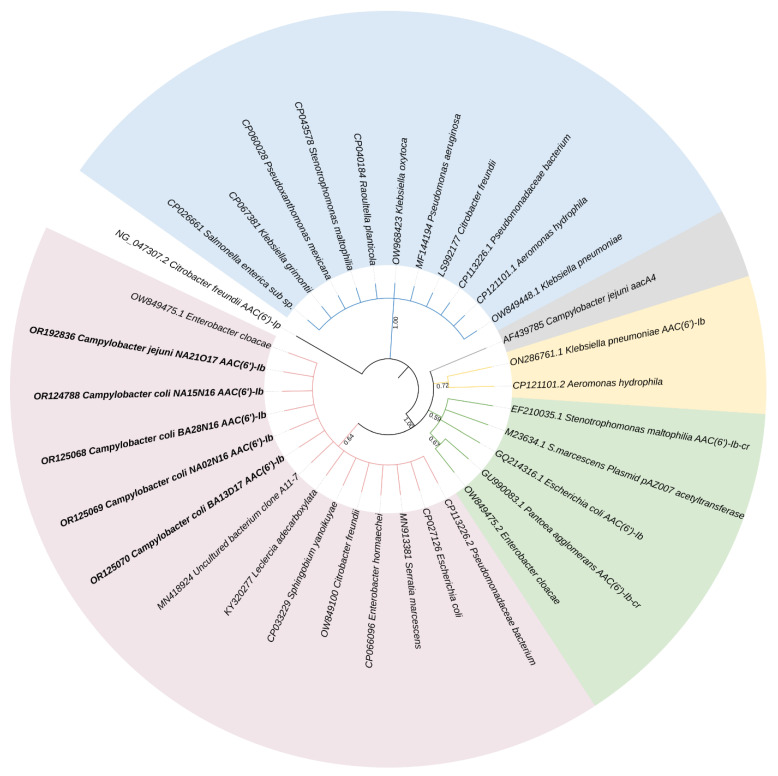
Molecular phylogenetic analysis by maximum likelihood method. The evolutionary history was inferred by using the maximum likelihood method based on the Kimura two-parameter model. The tree with the highest log likelihood (−1721.35) is shown. The percentage of trees in which the associated taxa clustered together is shown next to the branches. Initial tree(s) for the heuristic search were obtained automatically by applying Neighbor-Join and BioNJ algorithms to a matrix of pairwise distances estimated using the maximum composite likelihood (MCL) approach, and then selecting the topology with superior log likelihood value. A discrete Gamma distribution was used to model evolutionary rate differences among sites (five5 categories (+G, parameter = 200.0000)). The rate variation model allowed for some sites to be evolutionarily invariable ([+I], 8.19% sites). The tree is drawn to scale, with branch lengths measured in the number of substitutions per site. The analysis involved 34 nucleotide sequences. All positions containing gaps and missing data were eliminated. There were a total of 421 positions in the final dataset. Evolutionary analyses were conducted in MEGA7. ITOL version 6 was used to create the phylogenetic tree. The strains in our investigation are described using their bolded accession number code.

**Table 1 ijms-24-16116-t001:** Percentage of resistant *Campylobacter* isolates.

Antibiotics	*C. jejuni* (n = 91) % (n)	*C. coli* (n = 41) % (n)	Total (n = 132) % (n)
**Gentamicin**	14.3 (13)	9.8 (4)	12.9 (17)
**Kanamycin**	19.78 (18)	46.34 * (19)	28.03 (37)
**Tobramycin**	19.78 (18)	60.97 * (25)	58.33 (77)
**Streptomycin**	34.06 (31)	80.48 (33)	48.48 (64)
**Ciprofloxacin**	98.9 (90)	100 (41)	99.2 (131)
**Nalidixic Acid**	57.1 * (52)	22 (9)	46.2 (61)

* Significantly higher resistance (*p* < 0.05) of *C. jejuni* compared with *C. coli*.

## Data Availability

The statistical data used to support the findings of this study are available from the corresponding author upon request.

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
