# Peer review of "First Report of aac(6′)-Ib and aac(6′)-Ib-cr Variant Genes Associated with Mutations in gyrA Encoded Fluoroquinolone Resistance in Avian Campylobacter coli Strains Collected in Tunisia"

_ijms, 2023, doi:10.3390/ijms242216116_

Round 1

Reviewer 1 Report

Comments and Suggestions for Authors

Gharbi et al. performed a study to present the characteristics of campylobacter isolates in Tunisia, from poultry origin, harboring the aac (6')-Ib-cr variant gene . It seems that quinolone/fluoroquinolone resistance is established,  by chromosomal mutation in gyrA gene and evolve to aquire the aac(6')-Ib-cr in a genetic background already rich with the wild aac(6')-Ib gene. These genes were present in Campylobacter isolates exhibiting resistance to multiple antibiotics which restricts the range of available treatments.

Comments:

The manuscript was well organized and presented.  I have only one major remark – in my opinion, two or three sentences on the modes of spread of these genes, within the will greatly contribute to the quality of the work.

Author Response

*Comment: The manuscript was well organized and presented.  I have only one major remark – in my opinion, two or three sentences on the modes of spread of these genes, within the will greatly contribute to the quality of the work.

*Response: Dear reviewer, thank you for your positive evaluation to our work. Concerning your comment, in general it is well known that the spread of genes encoding antimicrobial-resistnce is principally mediated by the transfer of plasmids, transposons and integrons, sometimes but rarely by natural transformation and by phages. To highlight this, we added this sentence in the introduction section (highlighted by yellow color): ‘Globally, this enzyme is widely reported, either plasmid or chromosomally encoded, from Gram-negative bacteria especially in Enterobacteriaceae of human, farm animal and environmental origins, showing its great potential to be disseminated [10]. In general, in bacteria, the spread of genes encoding antimicrobial-resistance is mediated mobile genetic elements such as plasmids, transposons and insertion sequences as well as integrons and rarely by phages, which enhance their rapid and large scale spread [11].’

Reviewer 2 Report

Comments and Suggestions for Authors

two issues that would widen the appeal of your study:

1. It is not clear how the samples were selected/ chosen. What was the prevalence of the isolates in broiler hens, and on what basis were the isolates sampled and selected? 

2. What is the significance of this in clinical terms? 

Author Response

* Comments :

  1. It is not clear how the samples were selected/ chosen. What was the prevalence of the isolates in broiler hens, and on what basis were the isolates sampled and selected? 

Response: As we mentioned in the section ‘Materials and Methods’ (4.2. Campylobacter Isolates Collection and Phenotypic Susceptibility) the studied strains were from our previous collection, in article about them was published and indicated in the manuscript (reference 12) (Gharbi et al. Relationships between virulence genes and antibiotic resistance phenotypes/genotypes in Campylobacter spp. isolated from layer hens and eggs in the north of Tunisia: Statistical and computational insights. Foods. 2022;11:3554). In that article we described the protocols of collection and identification of the isolates. In the previous study isolates were not investigated for the presence of the aac(6’)-Ib-cr gene.

  1. What is the significance of this in clinical terms?

Response: Fluoroquinolones are among the most important antibiotics used for the treatment of campylobacter infections and sometimes also aminosides (aminoglycosides) are used. Therefore, resistance to both families by the studied enzyme AAC(6’)-Ib-cr, significantly reduce choices to treat these infections especially when infectious Campylobacter strains are also resistant to macrolides and tetracycline, which was the case for the majority of our strains.  We already highlighted this in the introduction section, such as the sentence at the end of the introduction ‘Fluoroquinolones are among the three most widely used antimicrobials (macrolides and tetracyclines) used to treat systemic infection caused by Campylobacter; in addition, gentamicin (an aminoglycoside) is also used in some cases [1,2]…... Taken together, it is crucial to identify the strains harboring the aac (6')-Ib-cr variant in order to employ fluoroquinolone antimicrobials appropriately in human therapy’ AND in other part of the discussion section.

Reviewer 3 Report

Comments and Suggestions for Authors

This study investigates the aac(6′)-Ib and aac(6')-Ib-cr variant gene associated with mutations in gyrA encoded fluoroquinolone resistance in avian Campylobacter coli strains collected in Tunisia. The study is well-designed and executed, and the findings are important and relevant.

Overall, the study provides valuable information about the prevalence and distribution of the aac(6')-Ib and aac(6')-Ib-cr genes in Campylobacter isolates from Tunisia. However, further research is needed to investigate the clinical significance of these genes and the mechanisms by which they confer resistance to antibiotics.

Major comments:

1-The sample size is relatively small, especially for C. jejuni.

2-The study only looked at two species of Campylobacter (C. jejuni and C. coli). Other species of Campylobacter may also be capable of harboring the aac(6')-Ib and aac(6')-Ib-cr genes.

3- The study did not investigate the mechanisms by which the aac(6')-Ib-cr gene confers resistance to fluoroquinolones. This is an important area of future research, as it could lead to the development of new strategies to combat antibiotic resistance in Campylobacter.

The introduction is well-written and informative. It provides a good overview of the background of the study, including the importance of Campylobacter as a zoonotic pathogen, the use of fluoroquinolones to treat Campylobacter infections, and the emergence of antimicrobial resistance. The introduction also clearly states the purpose of the study, which is to investigate the presence of the aac(6')-Ib-cr variant in Campylobacter isolates from poultry origin in Tunisia.

My only comment is that the introduction is a bit long and could be shortened by removing some of the background information that is not essential to the study.

In the methods section: What is the breakpoint values used for each antibiotic to define resistance. This would allow readers to compare your results to those of other studies.

You mentioned that you used the boiling method to extract DNA. You could provide a brief description of this method.

The results section is well-written and informative.  Here are a few minor suggestions for improvement:

In the first sentence of the first paragraph, you could replace "approximately" with "nearly."

In the second sentence of the first paragraph, you could replace "by MAMA-PCR" with "using MAMA-PCR."

In the third sentence of the first paragraph, you could replace "found in the one" with "found in one."

In the first sentence of the second paragraph, you could replace "showed that 31 out of 41 C. coli (75.60%) isolates and one out of 91 C. jejuni (0.98%) isolates were positive for the aac(6’)-Ib gene" with "showed that 31/41 (75.60%) C. coli isolates and 1/91 (0.98%) C. jejuni isolates were positive for the aac(6')-Ib gene."

In the second sentence of the second paragraph, you could replace "revealed two fragments (272 bp and 210 bp) of DNA on gel electrophoresis" with "revealed two DNA fragments (272 and 210 bp) on gel electrophoresis."

In the third sentence of the second paragraph, you could replace "when treated by restriction enzyme BtsCI indicating the occurrence of both the wild aac(6')-Ib gene and the aac(6')-Ib-cr variant" with "when treated with restriction enzyme BtsCI, indicating the occurrence of both the wild-type aac(6')-Ib gene and the aac(6')-Ib-cr variant."

In the second sentence of the third paragraph, you could replace "and based on the results of antibi-otic resistance profiles and on products digested with BtsCI" with "based on their antibiotic resistance profiles and BtsCI digestion results."

In the third sentence of the third paragraph, you could replace "The se-quences of aac(6´)-Ib gene were deposed in GenBank under accession numbers" with "The sequences of the aac(6')-Ib gene were deposited in GenBank under accession numbers."

In the fourth sentence of the third paragraph, you could replace "The sequence analyses showed that Tuni-sian aac(6´)-Ib sequences found in this study (OR124788, OR125068, OR125070, 101 OR125069, and OR192836), were closely related to other sequences from various species 102 such as Enterobacter cloacae, Citrobacter freundii, and Pseudomonadaceae" with "The sequence analysis showed that Tunisian aac(6')-Ib sequences found in this study (OR124788, OR125068, OR125070, OR125069, and OR192836) were closely related to other sequences from various species, such as Enterobacter cloacae, Citrobacter freundii, and Pseudomonadaceae."

In the fifth sentence of the third paragraph, you could replace "the phyloge-netic branch was supported by a high bootstrap value of 99%" with "the phylogenetic branch was supported by a high bootstrap value of 99%."

In the sixth sentence of the third paragraph, you could replace "clustering using Maximum Likelihood method defined four sequence clusters and one singleton corresponding to C. jejuni AF4339785 (aacA4)" with "clustering using the Maximum Likelihood method defined four sequence clusters and one singleton corresponding to C. jejuni AF4339785 (aacA4)."

In the seventh sentence of the third paragraph, you could replace "Different clusters can ex-hibit similar properties while others show substantial differences in their characteristics" with "Different clusters can exhibit similar properties, while others show substantial differences in their characteristics."

Author Response

*Comments:

  1. This study investigates the aac(6′)-Ib and aac(6')-Ib-cr variant gene associated with mutations in gyrA encoded fluoroquinolone resistance in avian Campylobacter coli strains collected in Tunisia. The study is well-designed and executed, and the findings are important and relevant.

Overall, the study provides valuable information about the prevalence and distribution of the aac(6')-Ib and aac(6')-Ib-cr genes in Campylobacter isolates from Tunisia. However, further research is needed to investigate the clinical significance of these genes and the mechanisms by which they confer resistance to antibiotics.

Response: Dear reviewer thanks you for your positive evaluation to our article. We will respond to your comments point by point and corrections are made in yellow color in the main manuscript.

  1. Major comments:

1-The sample size is relatively small, especially for C. jejuni.

Response: These strains were previously collected and were mainly reported in our previous work (reference 10 in the article). We do not exaggerate if we tell that we are the only team in Tunisia that isolated and characterized Campylobacter isolate, this was began few years ago, and we have only these strains. Surely, characterization of more Campylobacter strains will provide more valuable data about the epidemiology of Campylobacter in Tunisia. Our team is going to collect and characterize more isolate in the near Future.

2-The study only looked at two species of Campylobacter (C. jejuni and C. coli). Other species of Campylobacter may also be capable of harboring the aac(6')-Ib and aac(6')-Ib-cr genes.

Response: as mentioned above, the collected isolates belong only to C. jejuni and C. coli species. This might be linked that both species are dominant in the gastrointestinal tract of chickens.

3- The study did not investigate the mechanisms by which the aac(6')-Ib-cr gene confers resistance to fluoroquinolones. This is an important area of future research, as it could lead to the development of new strategies to combat antibiotic resistance in Campylobacter.

Response: You have reason about this, as bacteriologist not chemist specialist we focused only on the occurrence of this gene. I believe that several studies have investigated the chemical action of this enzyme on ciprofloxacin and norfloxacin. Surely, modification of the molecular structures will prevent the action of AAC(6’)-Ib.cr

4- The introduction is well-written and informative. It provides a good overview of the background of the study, including the importance of Campylobacter as a zoonotic pathogen, the use of fluoroquinolones to treat Campylobacter infections, and the emergence of antimicrobial resistance. The introduction also clearly states the purpose of the study, which is to investigate the presence of the aac(6')-Ib-cr variant in Campylobacter isolates from poultry origin in Tunisia.

  • My only comment is that the introduction is a bit long and could be shortened by removing some of the background information that is not essential to the study.

Response: As you told, we tried to highlight all the background of the study and all parts are necessary to readers in order to understand the article. I just can delete this sentence that was before the aim of the study: ‘In spite of the fact that PMQR genes typically only impart low-level resistance to fluoroquinolones without concurrent resistance to quinolones, they provide a genetic background for the selection of chromosomal mechanisms leading to increased resistance levels in addition to hindering therapy.’

  • In the methods section: What is the breakpoint values used for each antibiotic to define resistance. This would allow readers to compare your results to those of other studies.

Response: we modified this sentence as follows: ‘…Interpretation of antimicrobial susceptibility against each antibiotic was determined following the breakpoint values of the European Committee on Antimicrobial Susceptibility Testing (EUCAST) [35, 36].’  

  • You mentioned that you used the boiling method to extract DNA. You could provide a brief description of this method.

Response: ‘The genomic DNA of collected isolates was extracted using the boiling method [32]. Briefly, Campylobacter isolates were grown in 2 mL Bolton broth and plated on Karmali agar. Campylobacter colonies were then harvested and suspended in 100 μL TE buffer (10 mM Tris, 1 mM EDTA, pH 8.0). Cell suspensions were heated at 100 °C for 10 min and then cooled at -20 C°. Thereafter, cell suspensions were centrifuged at 8000 rpm for 5 min [32]. The supernatant containing DNA was collected, transferred into a new tube, and then stored at −20 °C until use.

5- The results section is well-written and informative.  Here are a few minor suggestions for improvement:

-In the first sentence of the first paragraph, you could replace "approximately" with "nearly."

Response: This was corrected as follows: ‘Nearly 90% of C. jejuni and 80% of C. coli…………’

  • In the second sentence of the first paragraph, you could replace "by MAMA-PCR" with "using MAMA-PCR."

Response: this was corrected (..the Thr-86-Ile amino acid substitution in their GyrA as determined using MAMA-PCR)’

  • In the third sentence of the first paragraph, you could replace "found in the one" with "found in one."

Response: this was corrected

  • In the first sentence of the second paragraph, you could replace "showed that 31 out of 41 C. coli (75.60%) isolates and one out of 91 C. jejuni (0.98%) isolates were positive for the aac(6’)-Ib gene" with "showed that 31/41 (75.60%) C. coli isolates and 1/91 (0.98%) C. jejuni isolates were positive for the aac(6')-Ib gene."

Response: this was corrected

  • In the second sentence of the second paragraph, you could replace "revealed two fragments (272 bp and 210 bp) of DNA on gel electrophoresis" with "revealed two DNA fragments (272 and 210 bp) on gel electrophoresis."

Response: this was corrected

  • In the third sentence of the second paragraph, you could replace "when treated by restriction enzyme BtsCI indicating the occurrence of both the wild aac(6')-Ib gene and the aac(6')-Ib-cr variant" with "when treated with restriction enzyme BtsCI, indicating the occurrence of both the wild-type aac(6')-Ib gene and the aac(6')-Ib-cr variant."

Response: this was corrected

  • In the second sentence of the third paragraph, you could replace "and based on the results of antibi-otic resistance profiles and on products digested with BtsCI" with "based on their antibiotic resistance profiles and BtsCI digestion results."

Response: this was corrected

  • In the third sentence of the third paragraph, you could replace "The se-quences of aac(6´)-Ib gene were deposed in GenBank under accession numbers" with "The sequences of the aac(6')-Ib gene were deposited in GenBank under accession numbers."

Response: this was corrected

  • In the fourth sentence of the third paragraph, you could replace "The sequence analyses showed that Tuni-sian aac(6´)-Ib sequences found in this study (OR124788, OR125068, OR125070, 101 OR125069, and OR192836), were closely related to other sequences from various species 102 such as Enterobacter cloacae, Citrobacter freundii, and Pseudomonadaceae" with "The sequence analysis showed that Tunisian aac(6')-Ib sequences found in this study (OR124788, OR125068, OR125070, OR125069, and OR192836) were closely related to other sequences from various species, such as Enterobacter cloacae, Citrobacter freundii, and Pseudomonadaceae."

Response: this was corrected

  • In the fifth sentence of the third paragraph, you could replace "the phyloge-netic branch was supported by a high bootstrap value of 99%" with "the phylogenetic branch was supported by a high bootstrap value of 99%."

Response: this was corrected

  • In the sixth sentence of the third paragraph, you could replace "clustering using Maximum Likelihood method defined four sequence clusters and one singleton corresponding to C. jejuni AF4339785 (aacA4)" with "clustering using the Maximum Likelihood method defined four sequence clusters and one singleton corresponding to C. jejuni AF4339785 (aacA4)."

Response: this was corrected

  • In the seventh sentence of the third paragraph, you could replace "Different clusters can ex-hibit similar properties while others show substantial differences in their characteristics" with "Different clusters can exhibit similar properties, while others show substantial differences in their characteristics."

Response: this was corrected